# Probabilistic Modelling for Unsupervised Analysis of Human Behaviour in Smart Cities

**DOI:** 10.3390/s20030784

**Published:** 2020-01-31

**Authors:** Yazan Qarout, Yordan P. Raykov, Max A. Little

**Affiliations:** 1Department of Mathematics, Aston University, Birmingham B4 7ET, UK; yordan.raykov@gmail.com; 2Department of Computer Science, University of Birmingham, Birmingham B15 2TT, UK; maxl@mit.edu

**Keywords:** time series, probabilistic modelling, trajectory analysis, smart city

## Abstract

The growth of urban areas in recent years has motivated a large amount of new sensor applications in smart cities. At the centre of many new applications stands the goal of gaining insights into human activity. Scalable monitoring of urban environments can facilitate better informed city planning, efficient security, regular transport and commerce. A large part of monitoring capabilities have already been deployed; however, most rely on expensive motion imagery and privacy invading video cameras. It is possible to use a low-cost sensor alternative, which enables deep understanding of population behaviour such as the Global Positioning System (GPS) data. However, the automated analysis of such low dimensional sensor data, requires new flexible and structured techniques that can describe the generative distribution and time dynamics of the observation data, while accounting for external contextual influences such as time of day or the difference between weekend/weekday trends. In this paper, we propose a novel time series analysis technique that allows for multiple different transition matrices depending on the data’s contextual realisations all following shared adaptive observational models that govern the global distribution of the data given a latent sequence. The proposed approach, which we name Adaptive Input Hidden Markov model (AI-HMM) is tested on two datasets from different sensor types: GPS trajectories of taxis and derived vehicle counts in populated areas. We demonstrate that our model can group different categories of behavioural trends and identify time specific anomalies.

## 1. Introduction

The rising population in modern cities introduces many challenges in urban city planning including problems associated with improving the inhabitants’ quality of life and security. The utilisation of sensors and communication technology gives rise to the concept of Smart Cities and enables the adequate study of specific focus problems in urban development. Among the many examples are applications on structural health monitoring [1], where material conditions of civil infrastructure are monitored and automatically analysed to ensure population safety; smart waste management [2] to improve service provisioning by the utilisation of smart garbage bins and Internet of Things (IoT) technology; smart lighting solutions [3] for cost and greenhouse emission reduction by the installation of sensors and weather analytics on city streets; and, in the manufacturing industry [4], combining IOT with machine learning. With the growing interest and importance of sensor application in smart cities, there has also been a rise in the interest in human movement behavioural characteristics from sensors to identify movement trends and improve the understanding of traffic. Examples from recent literature on applications of human behavioural understanding include De Marsico et al. [5] user gait identification for automatic activation of controlled entry spaces, and Paul et al. [6], Torre-Bastida et al. [7] big data and social IOT was utilised for real time definition of human behaviours. For traffic analysis, Bhatti et al. [8] introduced an IOT accident detection and notification system that relies on multiple, commonly available smart-phone sensors.

The analysis of car traffic can aid the understanding of population dynamics in smart cities. This, on the other hand, leads to better capabilities for congestion control and accident detection Ahmad et al. [9]. For example, Kanungo et al. [10] studied traffic light management using video surveillance analysis, whereas Ozkurt and Camci [11] use surveillance footage to estimate density patterns. This focus on video surveillance data has prompted research on camera placement for greater area coverage [12] to reduce the high cost accumulation of such sensors. However, the high costs and privacy concerns related to CCTV data are a key driver of the search of alternative sensor resources: such as density counts, noise levels or automated geolocation data (such as GPS).

Despite the prior work on behaviour understanding from GPS [13,14,15], a lot of the GPS data analysis remains challenging due to noise, irregular sampling, heterogeneity and frequent interruption of data collection as environmental factors cause missing data [16]. To address this Ellam et al. [17], a recently proposed parameter estimation framework for spatial interaction models was demonstrated to simulate accurately the flow of customers. In scenarios where large amounts of individual-level data is available, we can also try to approach the problem reversely and first search for recurring patterns from large databases of movement in cities. Frequently sampled monitoring data such as the GPS data can be highly multimodal. To account for this, Witayangkurn et al. [18] proposed the use of a Hidden Markov Model (HMM) to detect anomalies from large scale GPS data. Nonetheless, this approach does not consider the time self-dependence of the observation sequence and requires that the model complexity be predefined which may bias the unsupervised approach to the analysis. The geospatial relationship between the journeys of individuals is often dependent on various contextual factors such as differences between weekend/weekday behaviour, peak/off-peak behaviour and individual idiosyncrasies.

To address some of these challenges, we design an exploratory nonparametric probabilistic modelling approach suitable for this type of data. We deploy a flexible HMM with augmented latent space to account for various types of contextual information. This contextual information can include difference between peak/off-peak time driving, weekday/weekend differences and person specific manoeuvrability decisions. To achieve this, we propose an intuitive Adaptive Input Hidden Markov Model (AI-HMM). Building on the theory of the Vector Autoregressive Hidden Markov Model (VAR-HMM) [19] to allow for autoregressive self-dependence between observations, we introduce a new discrete and independent semi-Markov variable which acts as a parent to the state indicator variable in the graphical model to represent environmental factors that influence movement behaviour. For example, in our case study in Section 5 on GPS data, we assume that the growing complexity of human trips, as more trajectories are observed, can be well captured with Bayesian nonparametric structure and that environmental factors such as time of day (e.g., peak or off-peak) may also influence the modality of the behaviour and therefore should be considered in the design of the generative model structure. We show that we are able to learn common patterns occurring across different contextual realisations, in addition to their specific generative distributions. To summarise, the contributions of this paper are as follows.

We develop an extension of the ubiquitous Hidden Markov model, which can robustly segment different behaviours in monitoring applications while accounting for present contextual factors.We have derived a principled framework for inference in the developed AI-HMM approach.We have shown the particularly suitability of the nonparametric switching autoregressive model for analysis of GPS and traffic count data.

This paper will continue as follows. Section 2 discusses related studies, Section 3 summarises the background of the used techniques, Section 4 explains the details of the proposed framework, Section 5 and Section 6 summarise the algorithm experiments and its results and Section 7 concludes the report.

## 2. Related Work

Human behaviour profiling and automated analysis has great potential for both security and retail commercial applications in smart city designs. However, labelled sensor data of such nature is usually scarce and unsupervised clustering or direct time series methods must be used [20,21,22]. One commonly used pipeline is the online segmentation of crowd data from video feeds. For example, Mehran et al. [14] proposed a novel Social Force model for identifying abnormal behaviours in crowds. The approach assumes that located moving particles on sequences of frames are individuals and uses them to estimate their interaction forces throughout time and estimate the “normal” behaviour of the crowd. Then, a bag-of-words classifier is trained to separate normal and abnormal crowd behaviours. Mahadevan et al. [23] also proposed a supervised anomaly detection method based on crowd videos. However, [23] evaluated their methods on a significantly smaller database of videos. Rodriguez et al. [24] developed a partially Bayesian approach to the same problem which allows for robust analysis of crowd videos completely unseen during training. While civic value can be demonstrated through the analysis of videos at crowded locations, such data provides only localised snapshots of the movement within the city. It is available only for a small set of chosen locations and requires a large amount of labelled location specific training data. Video feeds have also been used extensively for activity recognition applications [20,25,26]; however, such techniques are likely far from being applicable on larger spatial scale across a city.

Continuous location data can be used to study a more diverse geographic perspective on personal mobility and daily activity patterns. Jiang et al. [21] investigated how to cluster daily patterns of activity in the city using both GPS data from individuals as well as additional survey data concerning their true activities. The study used the Chicago Regional Household Travel Inventory (CRHTI) dataset [27], which is one of the few data sources with fine labelled information in addition to GPS data. GPS trajectory data was not explicitly modelled but only used to identify different location areas for the participants. Jiang et al. [21] first constructed a calender type array indicating which of nine activities (work, home, school, etc.) occurred. Principal Component Analysis (PCA) was then applied followed by K-means clustering to infer patterns of daily activity across different regions of the city. The inferred patterns were used to reveal socio-demographic information about the region. The results of the study are informative on human behaviour; however, if GPS information is included in the analysis, the complexity of the data may affect the effectiveness of the proposed framework.

Shih et al. [22] modelled different mobility patterns based on GPS data from the Geolife dataset [28] to classify trips of people diagnosed with Alzheimer’s disease. The basis for this is the common assumption that a symptom of Alzheimer’s patients is spatio-temporal confusion. The abnormal trajectories were predicted based on thresholding the similarity to the training set. Although the results showed up to 97% accuracy rate in detecting abnormal mobility patterns, there was no ground truth data, and explicitly, no real information on the Alzheimer state of each subject. Instead, external sequences were added to the data of some individuals to represent Alzheimer’s patients. This can add clear biases to the analysis and may indicate that the accuracy of the technique on more representative data will not mimic that of the presented study. More recently, Yao et al. [15] also proposed a novel behaviour recognition framework for GPS data, demonstrated on trajectories of ships. They used a Long-Short Term Memory (LSTM) autoencoder to infer the latent structure in features which were extracted from sliding windows of the trajectory time series. By passing the dataset through a trained network, the inferred embeddings were extracted and stored for each input time series. These embeddings were then clustered with K-means to identify different behavioural patterns in ship data. However, trajectories of ship movement contain significantly smaller variations across patterns and a constrained space of behaviours which can be explored. The LSTM autoencoder is likely to require multiple different behaviour sequences in order to group the trajectories in the latent space. Also, when focused on segmenting anomalous events, K-means is known to lead to misleading clustering [29].

## 3. Dynamic Time Series Modelling

### 3.1. Vector Autoregressive Model (VAR)

The Vector Autoregressive (VAR) model [30] is a widely used linear time series model for dynamic multidimensional data. An order *p* VAR assumes each observation in a sequence can be modelled as a linear combination of *p* previous points in the sequence and a stochastic term. The VAR model is also a parametric model of the spectral density. Unlike Fourier-based methods which require a windowing strategy, VARs are defined in the time domain and we can use *p* to control the number of spikes we assume in the power spectrum of the input. Let $Y=(y_{1},y_{2},\cdots ,y_{T})$ be a stationary time series where $y_{t}\in R^{D}={\left(y_{t}^{\left(1\right)},\cdots ,y_{t}^{\left(D\right)}\right)}^{T}$, a VAR(*p*) model will have the form of
(1)$$y_{t}=\Theta_{1}y_{t-1}+\cdots +\Theta_{p}y_{t-p}+\mu +\epsilon_{t},$$
where $\Theta_{i}$ is a (D×D) matrix of weights, μ is the mean of the sequence *Y* and $\epsilon_{t}$ is a (D×1) vector of white noise with zero mean and covariance matrix Σ. Using the matrix notation $Y_{p+1:T}=\left(y_{p+1},y_{p+2},\cdots ,y_{T}\right)$ we can then write the VAR in the following form,
(2)$$Y_{p+1:T}=\Theta \bar{Y}+E,$$
with $\Theta =(\mu ,\Theta_{1},\cdots ,\Theta_{p})$, $\bar{Y}={\left(1,Y_{p:T-1}^{T},\cdots ,Y_{1:T-p}^{T}\right)}^{T}$, 1 being a vector of ones with the same length as $Y_{p+1:T}$ and $E=(\epsilon_{p+1},\cdots ,\epsilon_{T})$. Using this formulation, the parameters of a VAR model may be estimated in few main ways depending on the adopted formalism: via maximum likelihood estimation, using the Yule–Walker equations and via Bayesian inference. Inference in Bayesian VAR models is practically identical to statistical inference in Bayesian linear regression. Assuming, for practical convenience, a conjugate prior over the VAR parameters, the joint prior over the parameters Θ and the covariance matrix Σ is a coupled Matrix-Normal Inverse-Wishart (MNIW) distribution, implied by the Gaussian likelihood of the outputs.
(3)$$\begin{array}{c} \Sigma ∼\mathrm{IW}(n_{0},S_{0}),\\ \Theta |\Sigma ∼\mathrm{MN}(M,\Sigma ,K),\\ Y_{p+1:T}|\bar{Y},\Theta ,\Sigma ∼\mathrm{MN}\left(\Theta \bar{Y},\Sigma ,I\right), \end{array}$$
where $n_{0}$ is the number of degrees of freedom; *S* is the scale matrix; M,Σ and $S_{0}$ are the mean, covariance of the rows and covariance of the columns of parameter Θ, respectively; and I is the identity matrix. The posterior of the VAR coefficients is then
(4)$$p\left(\Theta |Y,\Sigma \right)\propto p\left(Y|\bar{Y},\Theta ,\Sigma \right)p\left(\Theta |\Sigma \right).$$

To compute posterior of the VAR noise covariance, it is more efficient to use the marginal likelihood, instead giving
(5)$$\begin{array}{c} p(\Sigma |Y)\propto p(Y|\Sigma )p(\Sigma ),\\ p\left(Y|\Sigma \right)\propto \int_{\Theta }p\left(Y|\bar{Y},\Theta ,\Sigma \right)p\left(\Theta |\Sigma \right)d\Theta . \end{array}$$

### 3.2. Autoregressive Infinite Hidden Markov Model (AR-iHMM)

The infinite Hidden Markov Model (iHMM) also known as the Hierarchical Dirichlet Process HMM (HDP-HMM) is a Bayesian nonparametric approach to the HMM first proposed by Beal et al. [31] then formalised by Teh et al. [32]. The nonparametric formalisation tackled the limitations of the traditional HMM of needing to pre-specify the structure of the model allowing it to automatically infer in adaptive manner the complexity depending on the amount of data observed. Building on the iHMM, Fox [33] included linear AR dependence between the observation sequence to form the AR-iHMM (or also known as HDP-AR-HMM). Figure 1 depicts the Probabilistic Graphical Model (PGM) representation of the iHMM and the AR-iHMM.

Let $z=(z_{1},\cdots ,z_{T})$ denote the state indicator variables and $\pi_{k}$ be the state specific transition distribution for state *k* where $z_{t}∼\mathrm{Categorical}\left(\pi_{z_{t-1}}\right)$. The observations $y_{t}$ are conditionally independent given $z_{t}$ and ${\bar{Y}}_{t}={\left(y_{t-1}^{T},\cdots ,y_{t-p}^{T}\right)}^{T}$; each observation is modelled with emission distributions $y_{t}|z_{t},{\bar{Y}}_{t}∼f\left(\Theta_{z_{t}}{\bar{Y}}_{t}\right)$ where $\Theta_{z_{t}}$ are the emission parameters for state indicated by $z_{t}$. The complete data likelihood can then be written as
(6)$$\begin{array}{c} p\left(z,Y\right)=\prod_{t=p+1}^{T}p\left(z_{t}|z_{t-1}\right)p\left(y_{t}|z_{t},{\bar{Y}}_{t}\right),\\ p\left(z,Y\right)=\prod_{t=p+1}^{T}\sum_{k=1}^{K^{+}}\pi_{z_{t-1},k}f\left(y_{t};\Theta_{k},{\bar{Y}}_{t}\right). \end{array}$$
using $K^{+}$ to denote the inferred number of represented states.

The iHMM defines a set of random probability measures $G_{k}$ for each state representing the dynamic observations *Y* sampled from a Dirichlet Process (DP) with a global probability measure $G_{0}$ and concentration parameter α. The base probability measure itself is DP distributed with concentration parameter γ and a base probability measure *H*
(7)$$\begin{array}{c} G_{0}|\gamma ,H∼DP\left(\gamma ,H\right),\\ G_{k}|\alpha ,G_{0}∼DP\left(\alpha ,G_{0}\right), \end{array}$$
where measures $G_{k}$ are conditionally independent given $G_{0}$. Intuitively, Equation (Equation 7) means that random distributions $G_{k}$ vary around $G_{0}$ with variability governed by γ, whereas $G_{0}$ varies around the base distribution *H* with variability α. Including explicit hyperparameter controlling the self-transitions probability in the AR-iHMM [31,33] we can summarize the full AR-iHMM as follows,
(8)$$\begin{array}{c} \beta |\gamma ,H∼GEM(\gamma ),\\ \pi_{k}|\alpha ,\beta ∼DP\left(\alpha +\kappa ,\frac{\alpha \beta +\kappa \delta_{k}}{\alpha +\kappa }\right),\\ \Theta_{k}|H∼H,\\ z_{t}|z_{t-1},{\left(\pi_{k}\right)}_{k=1}^{\infty }∼\pi_{z_{t-1}},\\ y_{t}|z_{t},{\bar{Y}}_{t},{\left(\Theta_{k}\right)}_{k=1}^{\infty }∼f\left(\Theta_{z_{t}}{\bar{Y}}_{t}\right), \end{array}$$
where $\beta ={\left(\beta_{k}\right)}_{k=1}^{\infty }$ is both a mixing prior for the top level DP and a base measure for the lower level DP, GEM (which stands for Griffiths, Engen and McCloskey [34]) represents the stick breaking construction and ${\left(\cdot \right)}_{k=1}^{\infty }$ represents an infinite set.

## 4. Adaptive Input Infinite Hidden Markov Model (AI-iHMM)

### 4.1. Model Specification

AR-iHMMs are powerful models that are capable of modelling complex dynamic data effectively. However, as a generative model, they assume that the state indicator $z_{t}$ is only dependent on the state indicator at the previous time step $z_{t-1}$. The model also assumes that the full time series share the same upper layer mixing prior β and therefore the same set of transition distributions $\Pi ={\left(\pi_{1}^{T},\pi_{2}^{T},\cdots ,\pi_{K}^{T}\right)}^{T}$. In some real-world problems, there are independent and discrete features $\tau =(\tau_{1},\tau_{2},\cdots ,\tau_{T})$ that affect the model’s state indicators and transition distributions. The semi-Markov variable τ can represent certain events or periods for the dataset where data generated during any fixed realisation of $\tau_{t}=v$ can share the same mixing prior β and same set of transition distributions Π, but have different distributions for different realisations *v*. For example, in the case of human movement analysis, different periods of the day $\tau_{t}\in \left\{1,2\right\}$, such as off-peak and peak time, respectively, can have different behavioural traits with separate parameters $\left\{\Pi_{1},\Pi_{2}\right\}$ and $\left\{\beta_{1},\beta_{2}\right\}$. Although the desired separation can be achieved by modelling the data generated for every realisation separately with different models, it will likely result in assigning exclusive states and state emission parameters $\Theta_{k}$ for each model. This will raise the problem of identifying a suitable method to measure state overlap across different models, specifically in cases like human movement behaviour where any small change in the states parameters may be important.

To avoid this problem, we propose a novel Adaptive Input infinite Hidden Markov Model (AI-iHMM) where an additional DP layer is added to the models hierarchy allowing for the generation of a different base measure $G_{0}^{\left(v\right)}$ for each unique value of τ while still sharing the same upper level base distribution *H* for the sampling of the model parameters $\Theta_{k}$. Figure 2 depicts the PGM of the proposed model with the respective joint probability distribution and formalisation.
(9)$$\begin{array}{c} P\left(\tau ,z,Y\right)=\prod_{t=p+1}^{T}P\left(\tau_{t}\right)P\left(z_{t}|z_{t-1},\tau_{t}\right)P\left(y_{t}|z_{t},{\bar{Y}}_{t}\right), \end{array}$$
(10)$$\begin{array}{c} Q|\psi ,H∼DP(\psi ,H),\\ G_{0}^{\left(v\right)}|\gamma ,H∼DP\left(\gamma ,Q\right),\\ G_{k}^{\left(v\right)}|\alpha ,G_{0}^{\left(v\right)}∼DP\left(\alpha ,G_{0}^{\left(v\right)}\right), \end{array}$$
where *H* is the base probability measure for the upper layer DP with concentration parameter ψ, *Q* is the master probability measure for the middle layer DP associated with the possible values of τ with concentration parameter γ, $G_{0}^{\left(v\right)}$ is the global probability measure for the lower layer DP associated with $\tau_{t}=v$ with shared concentration parameter α across all values of *v*, and $G_{j}^{\left(v\right)}$ is the random probability measure for state *j* when $\tau_{t}=v$.

The possible values of τ can be inferred adaptively as will be described in Section 4.2 by placing an appropriate prior on them (for example the DP would be a conjugate choice) and estimate them using modified MCMC inference. This is useful when it is known that there is an underlying variable affecting z and the values for each time instance are unknown. For more intuitive analysis and better understanding of the effect the contextual category has on the transition dynamics of the time series data, τ can also be set as a fixed categorical input to the model. This allows for more precise inference of the transition dynamics associated with each context *v* and for their custom selection based on the problem of interest. Depending on the complexity of the data, the estimation of input τ in addition to the estimation of the remaining model parameters may be result in numerous local minima if no prior knowledge is known about the model. Therefore, it would be advantageous to train with partial knowledge of τ or pre-train with a fixed input when inferring it adaptively. In the general case of a completely adaptive model, using the stick breaking construction, the model may be written as
(11)$$\begin{array}{c} \lambda |\psi ,H∼GEM(\psi ),\\ \beta_{v}|\gamma ,\lambda ∼DP\left(\gamma ,\lambda \right),\\ \pi_{k}^{\left(v\right)}|\alpha ,\beta_{v}∼DP\left(\alpha +\kappa ,\frac{\alpha \beta_{v}+\kappa \delta_{k}}{\alpha +\kappa }\right),\\ \epsilon_{j,i}^{\left(v\right)}∼\pi_{j,i}^{\left(v\right)}Dir\left(\frac{\eta }{V}\right)\lambda_{z_{t-1}},\\ \Theta_{k}|H∼H,\\ \tau_{t}|z_{t},z_{t-1},\epsilon ∼\epsilon_{z_{t-1},z_{t}},\\ z_{t}|z_{t-1},\tau_{t}=v,{\left(\pi_{k}^{\left(v\right)}\right)}_{k=1}^{\infty }∼\pi_{z_{t-1}}^{\left(v\right)},\\ y_{t}|z_{t},{\bar{Y}}_{t},{\left(\Theta_{k}\right)}_{k=1}^{\infty }∼f\left(\Theta_{z_{t}}{\bar{Y}}_{t}\right), \end{array}$$
where λ is the upper level mixing parameter and the middle level base measure, η is the prior parameter of the distribution of τ and $\epsilon_{j,i}^{v}$ is the posterior probability of $\tau_{t}=v$ when $z_{t}=i$ and $z_{t-1}=j$.

### 4.2. Inference

Similarly to the AR-iHMM structure, the parameters of the AI-HMM can be inferred by blocked Gibbs sampling. A truncation level *L* is set to denote the maximum number of states expected expected for the time series modelling so that k∈{1,2,⋯,L}. Given previously set transition distributions $\left\{\Pi_{v}^{\left(n-1\right)}\right\}$, mixing parameters $\left\{\beta_{v}^{\left(n-1\right)}\right\}$ and emission distribution parameters $\left\{\Theta_{k}^{\left(n-1\right)}\right\}$ and assuming a normal distribution likelihood on the autoregressive observation sequence; the first step is to block sample the input contextual vector τ. Given its Markov blanket, the posterior of $\tau_{t}$ is
(12)$$p\left(\tau_{t}|z_{t},z_{t-1}\right)\propto p\left(z_{t}|\tau_{t},z_{t-1}\right)p\left(\tau_{t}\right)p\left(z_{t-1}\right).$$

Here, $p(z_{t}|\tau_{t},z_{t-1})$ is simply the transition matrix elements $\left\{\pi_{z_{t-1},z_{t}}^{\left(v\right)}\right\}$ across all *v*, $p(\tau_{t})$ is the prior $Dir(\frac{\eta }{V})$, as $\tau_{t}$ is assumed to be categorically distributed with a Dirichlet prior and $p(z_{t-1})$ is the global prior of $z_{t-1}$. By design, changes in the value of $\tau_{t}$ should be sparse and infrequent since they represent broad categories of transition dynamics and cover long contextual periods relative to the time series. For the case of $z_{t}=z_{t-1}=k$, it can be possible that $\left\{\epsilon_{k,k}^{\left(v\right)}\right\}$ is similar for multiple values of *v* due to the closeness in value between probabilities $\left\{\pi_{k,k}^{\left(v\right)}\right\}$. A change in the value of $\tau_{t}$ directly implies that the state for $z_{t}$ has changed as well, due to the hierarchy of the model. This means that effectively $\tau_{t}$ is only sampled on when $z_{t}\neq z_{t-1}$, which naturally enforces state persistence. To sample
(13)$$\begin{array}{c} \tau_{t}∼\sum_{v=1}^{V}\epsilon_{z_{t-1},z_{t}}^{\left(v\right)}\propto \sum_{v=1}^{V}\pi_{z_{t-1},z_{t}}^{\left(v\right)}Dir\left(\frac{\eta }{V}\right)\lambda_{z_{t-1}}\delta \left(\tau_{t},v\right)\;\mathrm{if}z_{t}\neq z_{t-1};\\ \tau_{t}=\tau_{t-1}\;\mathrm{if}z_{t}=z_{t-1}. \end{array}$$

The next set of parameters to sample are z. This is done using a variation of the forward-backward algorithm where the (1×L) backward message $m_{t,t-1}$ is calculated by
(14)$$m_{t,t-1}=m_{t+1,t}\Pi_{\tau_{t}}\sum_{k=1}^{L}N\left(y_{t};\Theta_{k}{\bar{Y}}_{t},\Sigma_{k}\right).$$

The forward message $f\left(y_{t}\right)=\left(f_{1}\left(y_{t}\right),\cdots ,f_{L}\left(y_{t}\right)\right)$ for observation $y_{t}$ is
(15)$$f\left(y_{t}\right)=N\left(y_{t};{\left\{\Theta_{k}{\bar{Y}}_{t},\Sigma_{k}\right\}}_{k=1}^{L}\right)\times m_{t+1,t},$$
where the first element of $m_{t+1,t}$ is equal to 1 and $N\left(y_{t};{\left\{\Theta_{k}{\bar{Y}}_{t},\Sigma_{k}\right\}}_{k=1}^{L}\right)=\left(N\left(y_{t};\Theta_{1}{\bar{Y}}_{t},\Sigma_{1}\right),\cdots ,N\left(y_{t};\Theta_{L}{\bar{Y}}_{t},\Sigma_{L}\right)\right)$. The indicator variable $z_{t}$ can then be sampled by
(16)$$z_{t}∼\sum_{k=1}^{L}f_{k}\left(y_{t}\right)\pi_{z_{t-1},k}^{\left(\tau_{t}\right)}\delta \left(z_{t},k\right).$$

To sample the mixing parameters $\beta_{v}$, the following auxiliary variables must be sampled all of shape (L×L): $M_{v}$ representing the count of transitions occurring due to sampling directly from the base probability; ${\bar{M}}_{v}$ representing the counts of new transitions unobserved previously before in the sequence z and $N^{\left(v\right)}$ representing the counts of transitioning from state *j* to state *k*. To estimate $M_{v}$, for each $\left(j,k\right)\in {\left\{1,\cdots ,L\right\}}^{v}$ set $m_{j,k}^{\left(v\right)}=0$ and for $n=1,\cdots ,n_{j,k}^{\left(v\right)}$ sample
(17)$$x∼Ber\left(\frac{\alpha \beta_{k}^{\left(v\right)}+\kappa \delta \left(j,k\right)}{n+\alpha \beta_{k}^{\left(v\right)}+\kappa \delta \left(j,k\right)}\right),$$
then increment *n* and if x=1 increment $m_{j,k}^{\left(v\right)}$. Then, for j∈{1,⋯,L}, estimate $W_{v}$ by
(18)$$\omega_{j}^{\left(v\right)}∼Binomial\left(m_{j,j}^{\left(v\right)},\frac{\kappa }{\kappa +\alpha \beta_{j}^{\left(v\right)}}\right),$$
where $\omega_{j}^{\left(v\right)}$ denotes of the override counts when the base probability distribution was used to draw self transitions for state *j*. This can in turn be used to estimate ${\bar{m}}_{j,k}^{\left(v\right)}$
(19)$${\bar{m}}_{j,k}^{\left(v\right)}=\left\{\begin{array}{cc} m_{j,k}^{\left(v\right)} & j\neq k;\\ m_{j,j}^{\left(v\right)}-\omega_{j}^{\left(v\right)} & j=k. \end{array}\right.$$

$\beta_{v}$ may now be sampled by
(20)$$\beta_{v}∼Dir\left({\bar{m}}_{.,1}^{\left(v\right)},\cdots ,{\bar{m}}_{.,L}^{\left(v\right)}\right),$$
where ${\bar{m}}_{.,j}^{\left(v\right)}$ denotes the $j^{th}$ column of ${\bar{M}}_{v}$. The mixing parameter may then be used to sample $\pi_{k}^{\left(v\right)}$
(21)$$\begin{array}{c} \pi_{k}^{\left(v\right)}∼Dir\left(\alpha \beta_{1}^{\left(v\right)}+n_{k,1},\cdots ,\alpha \beta_{k}^{\left(v\right)}+\kappa +n_{k,k},\cdots ,\alpha \beta_{L}^{\left(v\right)}+n_{k,L}\right), \end{array}$$
where the hyperparameter κ is added to the parameters for state *k* to enforce self transitions. The final step in the iterative optimisation procedure is to sample the state specific vector autoregressive parameters $\Theta_{k}$ and $\Sigma_{k}$. When using a MNIW prior as explained in Section 3.1, the parameters may be updated given the observations *Y* and state indicator variables z. For more details on the derivation of the update, please see Fox [33].
(22)$$\begin{array}{c} S_{\bar{Y}\bar{Y}}^{\left(k\right)}={\bar{Y}}^{\left(k\right)}{\bar{Y}}^{{\left(k\right)}^{T}}+K,\\ S_{Y\bar{Y}}^{\left(k\right)}=Y^{\left(k\right)}{\bar{Y}}^{{\left(k\right)}^{T}}+MK,\\ S_{YY}^{\left(k\right)}=Y^{\left(k\right)}Y^{{\left(k\right)}^{T}}+MKM^{T},\\ S_{Y|\bar{Y}}^{\left(k\right)}=S_{YY}^{\left(k\right)}+S_{Y\bar{Y}}^{\left(k\right)}S_{\bar{Y}\bar{Y}}^{-(k)}S_{Y\bar{Y}}^{{\left(k\right)}^{T}}, \end{array}$$
where $Y^{\left(k\right)}$ and ${\bar{Y}}^{\left(k\right)}$ are the observations and lag matrix of lagged observations, respectively, that are classified to state *k*. $S_{\bar{Y}\bar{Y}}^{\left(k\right)},S_{Y\bar{Y}}^{\left(k\right)},S_{YY}^{\left(k\right)}$ and $S_{Y|\bar{Y}}^{\left(k\right)}$ are the posterior parameters such that
(23)$$\begin{array}{c} \Sigma_{k}∼\mathrm{IW}\left(n_{0}+\sum_{v}c_{k}^{\left(v\right)},S_{Y|\bar{Y}}^{\left(k\right)}+S_{0}\right),\\ \Theta_{k}|\Sigma_{k}∼\mathrm{MN}\left(S_{Y\bar{Y}}^{\left(k\right)}S_{\bar{Y}\bar{Y}}^{-(k)},\Sigma_{k},S_{\bar{Y}\bar{Y}}^{\left(k\right)}\right), \end{array}$$
where $c_{k}^{\left(v\right)}$ is the count of the number of observations in state *k* through when $\tau_{t}=v$.

The AI-HMM is a generative model that can represent the observation data statistically with the optimised model parameters. Therefore, given a pre-trained model and initial starting points, it may be possible to sample new observations that resemble the dynamics of the training data. The parameters can even be altered manually to represent different types of behaviours and trends. This allows for generation of new custom datasets that can be used for research across different fields.

## 5. Modelling With Fixed Inputs

To demonstrate the effectiveness of the AI-HMM, we tested the model on two different datasets that represent human and traffic behaviour in smart city environments. The first was the Dodgers loop dataset [35] containing loop sensor data counting the number of vehicles that pass through the Glendale on-ramp for the 101 North Freeway in Los Angeles in 5 min (288 reading per day) intervals over a period of 25 weeks. In addition to the presence of weekday morning and afternoon traffic-peak trends in the dataset, a baseball stadium in the vicinity of the ramp allows for the observation of the post-game traffic rise. This allows us to study the performance of the model at identifying commonly repeating daily trends that may change depending on external contextual information (weekend/weekday trends) as well as studying the outlier detection capabilities with the spike in density counts due to sparsely occurring games at the nearby stadium. The dataset is simple and uniformly sampled with easily identifiable trends that can be detected by simple data visualisation. Results can be seen in Section 5.1. The second dataset was the T-drive dataset [13,36] which contains GPS trajectory data from more than 10,000 taxis collected over a period of 1 week within the city of Beijing, China. Analysing this dataset is more challenging as GPS data can be very noisy, sparse, irregularly sampled, heterogeneous and its data collection can be frequently interrupted due to environmental factors causing missing data (rarely missing at random). This type of data is complex and allows us to explore the effect of multiple contextual realisations on the observation data such as time of day and user identity. It also relates well traffic analysis, an important topic in smart city research. The AI-HMM identifies interesting patterns and results as seen in Section 5.2.

### 5.1. Dodgers Loop Sensor Dataset

The AI-HMM was used to model the complete dodgers loop sensor dataset. Figure 3 shows the count scatter for an average game day on a weekday and on a weekend. As is can be seen from the sharp peaks on the plots, games on weekdays often occur in the evening, whereas games on the weekends can occur at a similar time in the evening or, as depicted in the figure, they can occur in the afternoon.

For this experiment, the observation instances $y_{t}=\left(c,h\right)$ where *c* is the vehicle count for the 5 min interval at time *t* and *h* is the hour of day when the observation took place. As weekday and weekend traffic dynamics commonly differ in density flow, the input feature $\tau_{t}\in \left\{1,2\right\}$ where $\tau_{t}=1$ denotes weekdays and $\tau_{t}=2$ is assigned for weekends. This choice is supported by the assumption that traffic dynamics generally repeat on a daily basis for weekends and weekdays separately, enforcing the belief of having different transition behaviours for each in the generative model. The VAR order was set to p=288 being the number of observations in a day.

Figure 4 shows an example of the results for five days in one week. Looking at the weekday plot, it can be seen that the morning and afternoon peak periods were clustered into states 1 (blue) and 4 (green), respectively. The sharp peaks in the evening (~23:00) on some weekdays corresponding to traffic rises due to the baseball games were clustered into state 6 (brown). On the weekend, the evening game on Saturday was also clustered into state 6; however, the afternoon (~16:30) game on Sunday was clustered into state 7 (pink); a state which is not encountered on weekdays when $\tau_{t}=1$. This demonstrates the dynamics of the AI-HMM. States are shared across the structures relating to the different values of $\tau_{t}$, however, there may be differences when a set of observations following a certain behaviour trends occur exclusively for realisation *i*.

To compare the results, we use Principal Component Analysis (PCA) on the density counts of vehicles per day. Figure 5 shows the first 4 PCs plotted against time. It can be seen that PC 2 seems to capture the dynamics of the morning peak hours, whereas PC 4 captures the dynamics of the lighter weekend midday traffic. PC 3 Captures the trend of the evening game which commonly occurs on weekdays and Saturdays. These results inform us which orthogonal direction in the multidimensional data distribution we can expect to see these respective dynamics. However, this is specific to this data only, any additional day vectors added may alter the expected results. Moreover, using this technique alone, it is not possible to identify which days contain the identified trends and which days do not. Within the first four PCs, there was no PC that appears to identify the evening peaks nor the afternoon baseball games. All these shortcomings are solved with the AI-HMM approach demonstrating the effectiveness of the technique.

### 5.2. T-Drive Dataset

#### 5.2.1. Data Preprocessing

GPS data are often in the form of a collection of geolocation data points in the form of $X=(x_{1},x_{2},\cdots ,x_{T})$ where $x_{t}=\left(time,\lambda ,l\right)\in R^{3}$, λ is the longitude coordinate and *l* is the latitude coordinate. The altitude is also included in some data sources; however, it is neglected for the purpose of this paper since it is not always available. Given that the altitude is recorded in the dataset, the feature be added for analysis in the proposed technique with no changes to the structure. Extreme outliers relating to errors in the sensors measurement and very short trajectories (a trajectory consisting of only five data points for example) may be neglected since they do not generally hold much information about the journey.

Using only the trajectory time series *X*, the feature sequence *Y* is calculated to describe the trajectory path where $Y=(y_{1},y_{2},\cdots ,y_{T})$, $y_{t}=\left(\lambda ,l,h,\Delta v\right)\in R^{4}$, *h* is the hour of day and Δv is the average velocity given the distance travelled between two points Δd. The distance is calculated with the Haversine Equation [37]
(24)$$h\left(\frac{\Delta d_{t}}{r}\right)=h\left(\Delta \lambda \right)+\cos\left(\lambda_{t-1}\right)\cos\left(\lambda_{t}\right)h\left(\Delta \;\right),$$
where $h\left(\phi \right)=\sin^{2}\left(\frac{\phi }{2}\right)$ and r=6371km is the radius of the Earth assumed to be constant on all locations on its surface. Δv is then calculated by $\frac{\Delta d_{t}}{\Delta time_{t}}$. From the T-drive dataset, the recorder journeys of 10 taxis were selected randomly and modelled with the proposed AI-HMM. GPS data is often recorded as a sequence of geolocation observations $X=(x_{1},x_{2},\cdots ,x_{T})$, where $x_{t}\in R^{3}=\left(time,\lambda ,l\right)$, λ is the longitude coordinate and *l* is the latitude coordinate. The feature sequence *Y* is calculated to describe the trajectory path where $Y=(y_{1},y_{2},\cdots ,y_{T})$, $y_{t}\in R^{4}=\left(\lambda ,l,h,\Delta v\right)$, *h* is the hour of day and Δv is the average velocity given the distance travelled between two points as calculated by the Haversine Equation [37].

#### 5.2.2. AI-HMM Modelling with Predefined Input τ

Beijing is a busy city that is known for congested streets throughout most hours of the day [38]. However, the trajectories of trips in peak and off-peak times are likely to have very different latent dynamics. For this part of the case study, we assume that $\tau_{t}\in \left\{1,2\right\}$ can represent peak and off-peak traffic times respectively, and they are preset prior to modelling based on the time of day the trajectory point was recorded in. Due to the smoothness of the sampled GPS trajectories, we opted for a low VAR order of p=2. The hyperparameters γ,α and κ can be set experimentally by Bayesian model selection or by placing an updatable prior over them as seen in Fox et al. [33]. After running the experiment until convergence, the model identified six states corresponding to different movement behaviour as presented in Figure 6.

For comparison, a t-distributed Stochastic Neighbour Embedding (tSNE)-based technique [39] was applied following the methodology of Singleton [40]. However, due to the complexity of the data, the technique failed at identifying meaningful clusters; experimentally converging to the sample data average. We also compare our framework with a deep learning Gated Recurrent Unit (GRU) autoencoder embedding model with K-means clustering, similar to that proposed by Yao et al. [15]. To enable direct comparison, the number of clusters for the K-means algorithm was set to 6, following the number of states that were automatically identified with the nonparametric AI-HMM framework.

Figure 6 shows the results obtained from the AI-HMM and the GRU frameworks while Table 1 shows the median velocity $v_{median}$ interquartile range (IQR) velocity $v_{IQR}$, and the most prominent time ranges $h_{prominent}$ for each of the states using both techniques. The plots depict a reconstructed scatter map of the data where each point represents an observation with its corresponding value and time-step, whereas the colours represent which state the points belong to according to the legend. States 2 (orange) and 5 (purple) are mostly focused in the middle of the city with median velocities of 18 km/h and 19km/h, respectively. Their most prominent hours of day indicate that they represent the morning and afternoon peak periods when the roads are highly congested. As night approaches and traffic subsides, they are replaced by state 4 (red), which is also most prominent in the center of the city with a median velocity of 23 km/h. State 1 (blue) is focussed on the highway network where velocities are expectedly higher with a median velocity of 26 km/h. The observations belonging to state 3 (green) with a median velocity of 0 km/h are mostly distributed near Beijing capital international airport, where taxis often spend prolonged periods of time in stationary motion awaiting customers. As previously mentioned, GPS data can be very noisy with sporadic measurement errors causing possible location jumps. This noise was not filtered prior to analysis; however, the model was capable of identifying the noisy observations and separating from the rest of the data into state 6 (purple).

The GRU analysis was also successful at identifying the noisy observations and filtering them into states 3 (green) and 6 (brown); however, the remaining states have very similar median velocities. Inspection of state velocity histogram distributions indicates that the clustering values have been affected by long periods of stationarity (influenced by taxis at the airport). The distribution of the data points in Figure 6 also shows that there is minimal location based clustering since most of the states overlap between all regions of the city (e.g., city centre and highways). Therefore, the GRU was effective at identifying the most separated clusters; however, it identified more specific and less diverged behaviour patterns. The AI-HMM framework was capable of identifying meaningful states that represent behaviour trends combining location information, velocity and hour of day patterns, yielding more informative and representative results.

## 6. Adaptive Input

In cases where the contextual input τ is known or can be assigned to the data, the AI-HMM demonstrates highly informative unsupervised analysis results outperforming other commonly used time series sensor analysis techniques in the literature. However, it is often the case that τ is partially missing, and must therefore be inferred from the data adaptively. To demonstrate the performance of adaptively inferring the input vector τ given a pre-trained model, another experiment was conducted on the T-drive dataset where τ was used to indicate the taxis identity. The full data of two different taxis was extracted and separated into 75% training set and 25% test set. For the training data points, the input was set to $\tau_{t}\in \left\{1,2\right\}$ representing taxi 1 and taxi 2, respectively, and they were fit with the AI-HMM model to learn all remaining parameters and variables $\Theta_{k},\mu {,}_{k}\Sigma_{k},\Pi_{v},\beta_{v}$ and $z_{train}$. The aim for the test data was to infer the values of $z_{test}$ as well as $\tau_{test}$ using Equation (Equation 13) assuming they are unknown.

As would be expected, the transition distributions of the two different taxis given the training data showed some overlap in behaviour time dynamics with the addition of some differences. This firstly proves the hypothesis that sensor data belonging to different contextual factors (i.e., person identity in this example) follow different time dynamics in an HMM structure. Correctly identifying these transition distributions allows for more accurate model fitting; the framework will refer to a more specific transitions matrix representing the state transitions under the specified contextual representation during optimisation, generating more informative clustering results. For example, if transitions into state 4 representing late night, off-peak travel is absent from taxi 2’s behaviour dynamics, indicating that this taxi avoids late night fairs and prefers to work at different times of day, whereas taxi 1 may prefer to work in such less congested times.

The presence of differences in the transition dynamics allow for accurate estimation of τ given the trained parameters on unseen data as reflected by the results obtained from the test data run. Figure 7 shows the AI-HMMs capability of estimating the correct identity of the taxi using adaptive τ inference with an accuracy of 93%. The state transition behaviour of an entity can help discover its identity.

## 7. Conclusions

This paper demonstrates a novel Bayesian nonparametric model for the identification of human behavioural trends from smart city sensors. We address the unsupervised problem of analysing complex data sequences representing human movement by designing a generative PGM that permits for the adaptive contextual switching of transition dynamics. Results obtained from applying the model to complex real-world data demonstrated effectiveness in identifying different dynamic behavioural trends and noise filtering capabilities. We have motivated the value of the presented approach via piratical problems in monitoring and compared the results to examples of alternative tools such as RNNs. The presented framework allows for easy augmentation of existing systems with an adaptive factor which can estimate the effect of discrete contextual information. By adaptively estimating the input value given a pre-trained model, it is possible to accurately identify the contextual realisation which the data belongs to, such as predicting the identity of the individual that generated the data.

As of now, the model is only limited to low dimensional time-series data. However, future work will include broadening the model to scenarios where the observation model can be high dimensional and where multiple dependent contextual variables can be included.

## Figures and Tables

**Figure 1 sensors-20-00784-f001:**
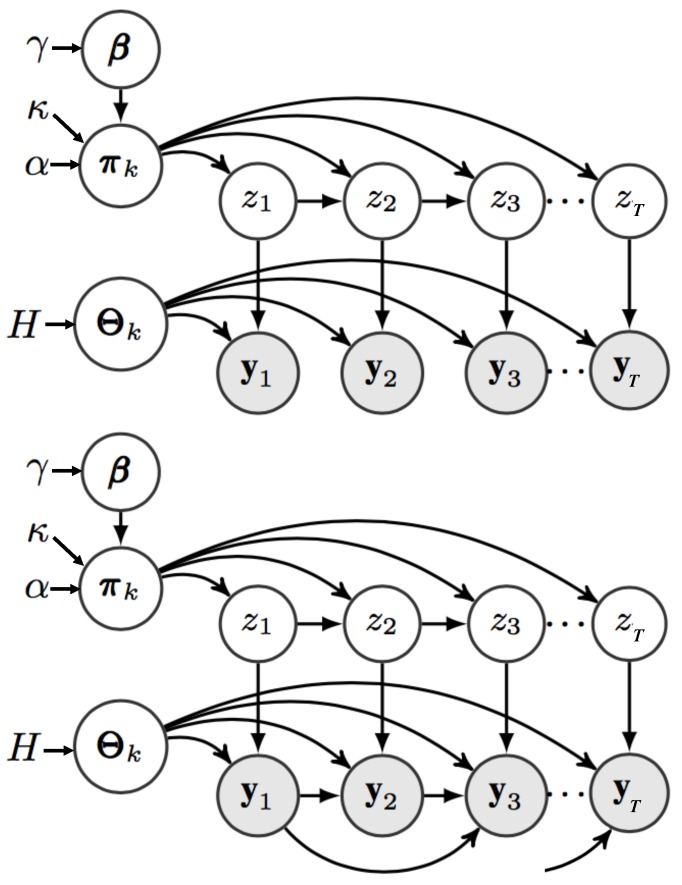
iHMM (top) and AR-iHMM (bottom) probabilistic graphical models. Mixing Parameters β are used to sample the transition distribution $\pi_{k}$, which the state indicators z are sampled from. The observations $y_{t}$ are generated from functions with parameters $\Theta_{z_{t}}$ which are in turn sampled from base distribution *H*. The AR-iHMM differs such that there are autoregressive dependence between observations.

**Figure 2 sensors-20-00784-f002:**
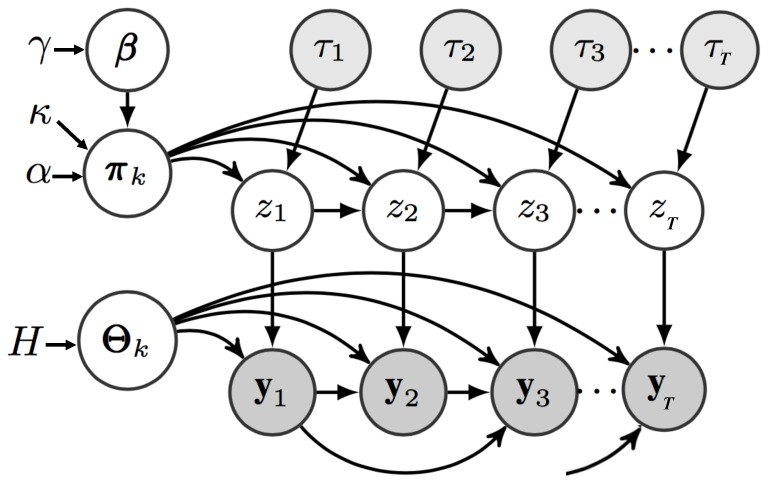
PGM of the AI-HMM structure. Observation $y_{t}$ being dependent on the indicator variable $z_{t}$ and *p* previous observations, whereas $z_{t}$ is dependent on $z_{t-1}$ in addition to the semi-Markov feature $\tau_{t}$. Parameters β are used to sample the transition distribution $\pi_{k}$ which the state indicators z are sampled from. The observations $y_{t}$ are generated from functions with parameters $\Theta_{z_{t}}$ which are in turn sampled from base distribution *H*.

**Figure 3 sensors-20-00784-f003:**
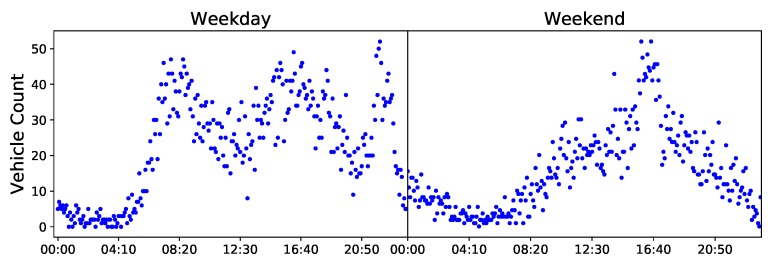
Scatter plots for vehicular density counts on an average game day on a weekday (**left**) and weekend (**right**). The first two distinct peaks on the weekday represent the morning and afternoon traffic peaks respectively, whereas the sharp peaks around 23:00 on the weekday and on 16:30 on the weekend represent traffic caused by fans leaving the stadium after a baseball game.

**Figure 4 sensors-20-00784-f004:**
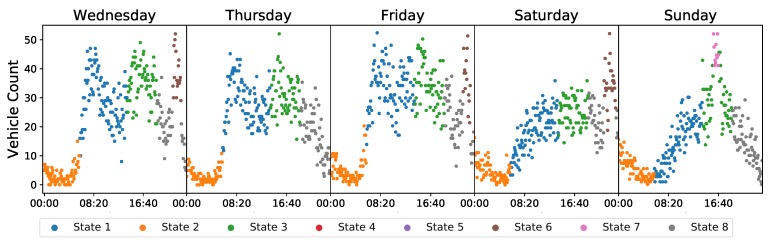
Scatter plots for vehicular density counts. Each colour represents a specific state that the respective observations was clustered into. Morning and evening traffic peaks on weekdays are clustered into state 1 (blue) and state 3 (green), respectively. Traffic caused by evening baseball games near 23:00 (Wednesday, Friday and Saturday) have been clustered into state 6 (brown), whereas traffic peaks caused by afternoon games at about 16:30 (Sunday) were clustered into state 7 (pink).

**Figure 5 sensors-20-00784-f005:**
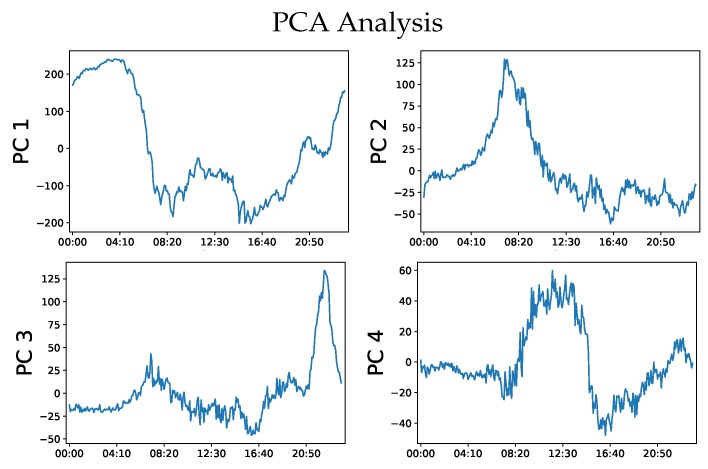
Figure depicting the first four (PCs with the largest eigen values in the PCA analysis of the Dodgers loop sensor dataset). PC 2 capture the dynamics of the morning peak; PC 3 represents the sharp traffic rise due to the evening baseball game, which commonly occurs on weekdays and Saturdays; and PC 4 captures the dynamics of the weekend light midday traffic.

**Figure 6 sensors-20-00784-f006:**
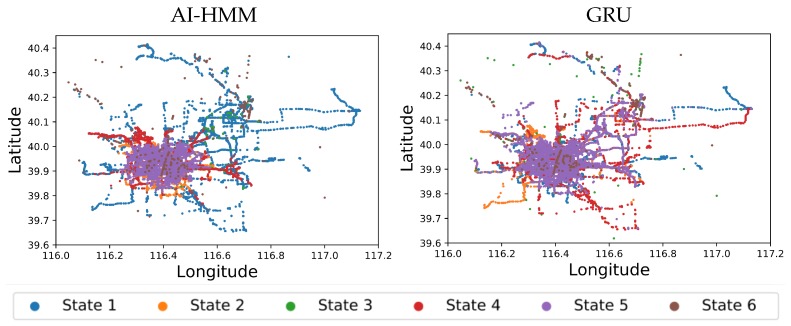
Results obtained from analysing 10 taxi trajectories using the novel AI-HMM framework (**left**) and the GRU technique (**right**). Each data point corresponds to a GPS observation where its colour represents the state to which it was assigned. The plots depict the longitude λ against latitude *l* reconstructions of the movements of the taxi locations on a map.

**Figure 7 sensors-20-00784-f007:**
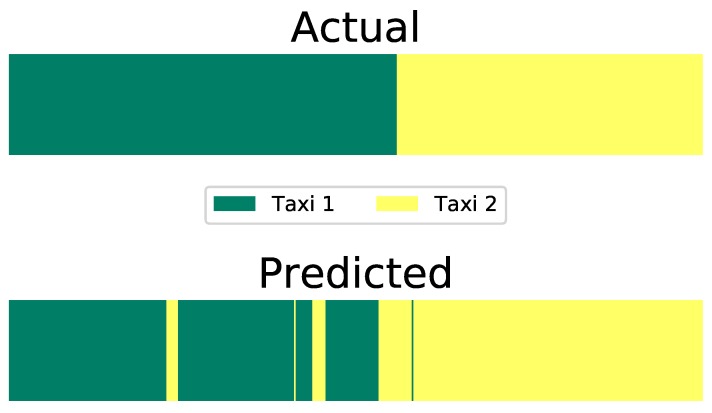
Prediction of the value of $\tau_{t}$ corresponding to the taxi identity on the test. The top plot shows the sample labels, whereas the bottom plot shows the predicted $\tau_{t}$ values using the AI-HMM with an accuracy of 93%.

**Table 1 sensors-20-00784-t001:** Cluster statistics (velocity in km/h).

State	AI-HMM			GRU		
	$v_{\mathrm{median}}$	$v_{\mathrm{IQR}}$	$h_{\mathrm{prominent}}$	$v_{\mathrm{median}}$	$v_{\mathrm{IQR}}$	$h_{\mathrm{prominent}}$
1	26	35	00:00–23:00	18	24	12:00–18:00
2	18	19	17:00–23:00	18	34	00:00–06:00
3	0	0	00:00–23:00	2147	7697	00:00–23:00
4	23	27	00:00–06:00	16	29	06:00–13:00
5	19	19	10:00–18:00	16	22	18:00–23:00
6	497	1336	00:00–23:00	186	176	00:00–23:00

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
