# Peer review of "Probabilistic Modelling for Unsupervised Analysis of Human Behaviour in Smart Cities"

_sensors, 2020, doi:10.3390/s20030784_

Round 1
Reviewer 1 Report
The paper is about the modelling of human motion in Smart cities through an unsupervised Markov Model able to capture features and cluster observations belonging to different domains: location, time, weekdays and weekends, etc.
The research method looks sound and the results are convincing, also, the study is timely and needed in the research community. Overall, my personal judgement as a reviewer is positive and I did not detect methodology-related and scientific flaws.
There are minor grammar mistakes that can be easily fixed by a read-through (e.g. "The for more details ..." page 9).
I would also put a couple of phrases about using the present methodology to generate new mobility datasets, as it would be of paramount importance for researchers.
Author Response
Thank you for your valuable feedback. We are happy that you appreciate the work and have taken your comments into accounts. The spelling and grammar mistakes have now been fixed. A paragraph has also been added on the possibility to generate new mobility datasets given a pre-trained model. Please refer to the red coloured paragraph that follows Equation 23. This is indeed a valuable addition to the paper and allows for interesting thought possibilities for future work.

Reviewer 2 Report
The author of the paper describes Probabilistic Modelling for Unsupervised Analysis of Human Behaviour in Smart Cities. Although the paper has sound knowledge about the smart cities based on the probabilistic approach. However, there are some information that need to addressed in the revised version. For instance,
The rationale about the proposed scheme is not clear. It is better to elaborate in the abstract and introduction. While adding in the Introduction, you must cite high quality papers published in reputable journals, for instance you can add below references to endorse your statements,
"Smartbuddy: defining human behaviors using big data analytics in social internet of things," in IEEE Wireless Communications, vol. 23, no. 5, pp. 68-74, October 2016.
doi: 10.1109/MWC.2016.7721744
Smart cyber society: Integration of capillary devices with high usability based on Cyber–Physical System, Future Generation Computer Systems Volume 56, March 2016, Pages 493-503.
"Big Data for transportation and mobility: recent advances, trends and challenges," in IET Intelligent Transport Systems, vol. 12, no. 8, pp. 742-755, 10 2018.
doi: 10.1049/iet-its.2018.5188.
"Automatic Non-Taxonomic Relation Extraction from Big Data in Smart City," in IEEE Access, vol. 6, pp. 74854-74864, 2018.
doi: 10.1109/ACCESS.2018.2881422
Secondly, include contribution and advantages in the end of the Introduction section.
Related work shall be revised by adding drawbacks for each reference. Current version seems like a summary of the existing papers.
I found that few equations, e.g. 7, 8, 11, and few other like these are not well explained. It is required to add details of the equations. e.g. how they are working in the proposed scheme.
Conclusion shall be revised and focus shall be giving to the findings in the results and discussions.
Author Response
Thank you for your valuable feedback. We have taken your comments into account and edited the paper accordingly. Specific answers to your comments are as follows:
Comment 1: The rationale about the proposed scheme is not clear. It is better to elaborate in the abstract and introduction. While adding in the Introduction, you must cite high quality papers published in reputable journals.
Response to Comment 1: Parts of the introduction and the abstract have been changed to clarify the reasoning behind the proposed scheme. We also cited some of the papers from these journals. Please refer to the red coloured parts in paragraphs one and two in the introduction section as well as the red coloured text in the abstract. Thank you.
Comment 2: Secondly, include contribution and advantages in the end of the Introduction section.
Response to Comment 2: An additional paragraph has also been added at the end of the introduction section to further highlight the contributions in bullet points. Again, added part is in red colour. Thank you.
Comment 3: Related work shall be revised by adding drawbacks for each reference. Current version seems like a summary of the existing papers.
Response to Comment 3: The related work was revised to mention the draw backs of the cited references where they were missing.
Comment 4: Conclusion shall be revised and focus shall be giving to the findings in the results and discussions.
Response to Comment 4: The conclusion was largely edited to focus on the findings and the results. Thank you.
Comment 5: I found that few equations, e.g. 7, 8, 11, and few other like these are not well explained. It is required to add details of the equations. e.g. how they are working in the proposed scheme.
Response to Comment 5: These equations are explained in more detail in some of the cited references such as the works by Fox et al. and Teh et al. They are well-known equations in the discipline and we feel that detailing them further may overflow the text in sections that should preferably remain concise and accurate. Thank you

Reviewer 3 Report
This research is very interesting, but wil be improve using a correct design of experiments and a multivariable analysis as in:
https://www.taylorfrancis.com/books/e/9780429454837/chapters/10.1201/9780429454837-16
and
https://www.taylorfrancis.com/books/e/9780429454837/chapters/10.1201/9780429454837-15
Is very important review specify implementation in a Smart City as in:
Maria De Marsico, Alessio Mecca, Silvio Barra:
Walking in a smart city: Investigating the gait stabilization effect for biometric recognition via wearable sensors. Computers & Electrical Engineering 80 (2019).
Fizzah Bhatti, Munam Ali Shah, Carsten Maple, Saif ul Islam:
A Novel Internet of Things-Enabled Accident Detection and Reporting System for Smart City Environments. Sensors 19(9): 2071 (2019)
Roberto A. Contreras-Massé, Alberto Ochoa-Zezzatti, Vicente García, José Mejía, Saúl González:
Application of IoT with haptics interface in the smart manufacturing industry. IJCOPI 10(2): 57-70 (2019)
Is important describe better the future research proposed.
Author Response
Thank you for your valuable feedback. We have taken your comments into account and edited the paper accordingly. The recommended references are useful and have mostly been used to back up some of the reasoning in the introduction so thank you for referring us to them. An additional paragraph has been added at the end of the conclusion section to discuss possible directions of future work which can include alternative experimentation design and analysis as recommended.

Reviewer 4 Report
The paper discusses two datasets in US and China for identifying human behavioural trends using sensors.
Broadly speaking, I would appreciate the authors to explain the link with a smart city.
Maybe I would be less technical and more exhaustive on the results (but I'm not an expert in statistics...)
Minor points
INTRO - l.68: section 6 seems missing; l.76 - check "behavior"; l. 181 - check "vehicles"
5- I would say WHY these 2 datasets have been chosen and something more about their characteristics (baseball is not exactly the more diffused sport all over the world...)
Author Response
Thank you for your valuable feedback. We have taken your comments into account and edited the paper accordingly. Specific answers to your comments are as follows,
Comment 1: Broadly speaking, I would appreciate the authors to explain the link with a smart city.
Response to Comment 1: We elaborated on the link with smart cities in the introduction section by adding some new relevant references. Thank you.
Comment 2: Maybe I would be less technical and more exhaustive on the results (but I'm not an expert in statistics...)
Response to Comment 2: The conclusion was largely edited to focus on the findings and the results. Thank you.
Comment 3: INTRO - l.68: section 6 seems missing; l.76 - check "behaviour"; l. 181 - check "vehicles"
Response to Comment 3: All checked and corrected where appropriate. Thank you.
Comment 4: I would say WHY these 2 datasets have been chosen and something more about their characteristics (baseball is not exactly the more diffused sport all over the world...)
Response to Comment 4: we further explained why these datasets have been selected, justified by their represented characteristics and relation to the research problem. Please refer to the red coloured text in Section 5. Thank you.
